# The efficacy of DPP IV inhibitors as adjunct therapy for patients with auto-immune Diabetes: A systematic review and meta-analysis

Laurette Nakhoul[1,2], Tracy Nakhoul[1,2], Maria Abi Azar[1,2], Frederic Harb[3], Nancy Fawzi Nakhoul[4*]

1 School of Medicine and Medical Sciences, Holy Spirit University of Kaslik, Jounieh, Lebanon,
2 Department of Endocrinology, Notre Dame des Secours University Hospital Center (NDSUH), Byblos, Lebanon, 3 Department of Biomedical Sciences, Faculty of Medicine and Medical Sciences, University of Balamand, 4 Department of Internal Medicine, Faculty of Medicine and Medical Sciences, University of Balamand, Kalhat, Tripoli, Lebanon

* nancy.nakhoul@balamand.edu.lb

## Abstract

### Rationale

Type 1 diabetes mellitus (T1DM) is characterized by autoimmune destruction of pancreatic β-cells, leading to insulin deficiency and hyperglycemia. Although insulin therapy remains the cornerstone of T1DM management, achieving optimal glycemic control remains challenging. Dipeptidyl peptidase-4 (DPP-4) inhibitors, approved for type 2 diabetes, enhance endogenous incretin action and may enhance β-cell function. Some clinical trials have explored their adjunctive use in T1DM. This systematic review and meta-analysis aimed to evaluate the efficacy and safety of DPP-4 inhibitors as an adjunct to insulin in patients with T1DM.

### Methods

We systematically searched PubMed, Cochrane Library, Medline (OVID), Scopus, and ClinicalTrials.gov up to January 2025 for eligible studies. Randomized controlled trials (RCTs) investigating DPP-4 inhibitors versus placebo, both on top of insulin therapy for at least 12 weeks in T1DM patients, were included. The primary outcome was the change in HbA1c. Secondary outcomes included blood glucose, C-peptide, insulin dosage, BMI, weight, adverse events, and HOMA2-β scores. Risk of bias was assessed using the Cochrane RoB 2.0 tool. Data were pooled using a random-effects model, with effect sizes expressed as mean differences (MD) and 95% confidence intervals (CI).

### Results

Out of 1,117 identified studies, seven RCTs comprising 333 participants (176 in the experimental group, 157 in the control group) were included. The addition of DPP-4

**Data availability statement:** All relevant data are within the paper and its Supporting Information files.

**Funding:** The author(s) received no specific funding for this work.

**Competing interests:** The authors have declared that no competing interests exist.

inhibitors did not result in a significant or sustained reduction in HbA1c overall, except for a transient improvement between 3 and 6 months (MD −0.10%, 95% CI −0.16 to −0.05, p = 0.0003). DPP-4 inhibitors significantly reduced daily insulin requirements, particularly bolus doses, and postprandial blood glucose (by −34.40 mg/dL), especially in patients with a BMI < 25 kg/m² and diabetes duration <3 years. No significant effects were observed on weight, BMI, fasting blood glucose, fasting or postprandial C-peptide beyond three months. HOMA2-β scores were significantly higher with DPP-4 inhibitors. Safety outcomes were comparable between groups.

## Conclusions

DPP-4 inhibitors appear safe as adjunct therapy to insulin in patients with T1DM. Although they do not offer sustained HbA1c reduction, they may reduce daily insulin requirements, improve postprandial glucose, and transiently enhance β-cell function. Further large-scale studies are needed to better define the subgroups that might benefit from this strategy.

## Registration

This study was registered in PROSPERO (CRD42024610965)

## Background

Diabetes is a metabolic disorder characterized by elevated blood glucose levels. It encompasses several types, including type 1 diabetes, which accounts for 5–10% of cases; type 2 diabetes mellitus, comprising 90–95% of cases and primarily caused by insulin resistance; and gestational diabetes [1]. Type 1 diabetes mellitus is an autoimmune disease, primarily affecting children, characterized by the destruction of pancreatic β-cells responsible for insulin production, resulting in insulin deficiency and subsequent hyperglycemia. [2]. The main treatment is insulin therapy; rapid and long-acting insulin injections or insulin pump; along with glucose monitoring through continuous glucose monitoring devices (CGM) or glucose strips [3]. Type 1 diabetes is difficult to control: patients need multiple insulin injections per day with considering insulin to carb ratio, carb counting and insulin sensitivity factor [4]. The mean HbA1c for type 1 diabetics ranges from 8.4% to 8.8% in those under 18 years of age, and from 7.4% to 8.5% in adults over 18 years. Around 25% of adults are controlled [5]. Many studies have addressed adjunct therapy to insulin to try to fill the unmet needs in those patients. Many small RCTs have looked at the effect of DPP-4 inhibitors, GLP-1 receptor agonists or SGLT-2 inhibitors as adjunct therapy to insulin for a better glycemic control although their use is not approved by any regulatory bodies like the FDA. [6–8]. The data and results are scarce and inconsistent. Therefore, the aim of this meta-analysis it to systematically summarize the evidence on adding DPP-4 inhibitors to insulin therapy in patients with type 1 diabetes.

Dipeptidyl Peptidase 4 or DPP-4 inhibitors are a class of drugs called incretins that is FDA approved for the treatment of type 2 diabetes. Glucagon-Like-Peptide-1 or GLP-1 is an incretin secreted in a glucose dependent manner by the L cells in the guts that stimulate insulin secretion when blood glucose levels rise. It also decreases glucagon secretion, caloric intake and delays gastric emptying. DPP-4 is an enzyme responsible for the inactivation of GLP-1 leading to a decrease in its half-life to one minute. Consequently, DPP-4 inhibitors will increase the half-life of endogenous GLP-1 in the blood stream and help regulate blood glucose levels [9].

Given that type 1 diabetes is more difficult to manage than type 2 diabetes, there is a need to explore adjunctive therapies to insulin that can improve glycemic control and reduce insulin requirements. DPP-4 inhibitors are one of the safest oral anti-diabetic medications. DPP-4 inhibitors are used in some clinical trials in type 1 diabetic patients (off-label use), and some have shown a better β-cell function [6,10]. Assessing the effect of DPP-4 inhibitors in this population on HbA1c mainly is primordial. HbA1c values require at least 3 months to change [11]. To the best of our knowledge, the last meta-analysis conducted on this topic was done 6 years ago and did not show a beneficial effect of DPP-4 inhibitors on HbA1c in type 1 diabetics, furthermore, the previous meta-analysis didn't exclude studies conducted for less than 12 weeks [12]. Thus, an updated systematic review and meta-analysis on DPP-4 inhibitors usage as an adjunct therapy on type 1 diabetes is necessary.

## Objectives

The objective of this study is to evaluate the effect of DPP-4 inhibitors, used as adjunct therapy with insulin, compared to placebo alongside insulin, on HbA1c, blood glucose levels, C-peptide levels, daily insulin dosage, BMI, weight change, and adverse events in patients with type 1 diabetes.

## Methods

We conducted this review following the Methodological Expectations for Cochrane Intervention Reviews and reported it following the Preferred Reporting Items for Systematic Reviews and Meta-Analyses (PRISMA) 2020 guidelines (*cf.* S1 Prisma 2020 Checklist).

### Types of studies

We included randomized controlled trials (RCTs) that evaluated the addition of any DPP-4 inhibitor to insulin therapy in patients with type 1 diabetes for a minimum duration of 12 weeks. Studies not published in English or unfinished trials were excluded from the analysis.

### Types of participants

Participants had to be type 1 diabetic patients regardless of the duration of diabetes and age and had to be treated with insulin only.

### Types of interventions

Patients in the intervention group were treated with any DPP-4 inhibitor—such as sitagliptin or saxagliptin—at any given dose, either alone or combined with other medications, alongside insulin therapy for at least 12 weeks. The control group received a placebo in addition to insulin for the same timeframe. Trials with a treatment period shorter than 12 weeks were not considered. In this study, we also compared outcomes between patients receiving sitagliptin (with or without other medications) plus insulin and those receiving placebo plus insulin, as well as sitagliptin alone versus placebo. Subgroup analyses were conducted according to treatment duration, diabetes duration, and body mass index (BMI).

## Outcome measures

The primary outcome of this study was the change in HbA1c levels following the addition of DPP-4 inhibitors to insulin therapy in patients with type 1 diabetes. Secondary outcomes included C-peptide levels, insulin dosage, blood glucose measurements, body mass index (BMI), weight, adverse events, and HOMA2-%B scores. Critical outcomes assessed were HbA1c (%), blood glucose (mg/dL), C-peptide (pmol/L), insulin dosage (units), BMI (kg/m²), weight (kg), and adverse events. For blood glucose, extracted data included fasting blood glucose, postprandial blood glucose, delta blood glucose (postprandial minus fasting), time in range, time in hyperglycemia, time in hypoglycemia, mean blood glucose, and glucose area under the curve (AUC). For C-peptide, data extracted included fasting C-peptide, postprandial C-peptide, delta C-peptide (postprandial minus fasting), and C-peptide AUC. Insulin dosage was assessed by daily insulin units, daily insulin units adjusted for weight (units/day/kg), basal insulin units, and bolus insulin units. Additionally, the HOMA2-%B score, reflecting β-cell function, was evaluated as an important secondary outcome.

## Search methods for identification of studies

**Electronic searches.** We systematically searched PubMed, the Cochrane Library, Medline (OVID), and Scopus for eligible articles, with the last search conducted in January 2025. No filters were applied regarding language or publication status. The detailed search strategy, including keywords used, is provided in the S2 Appendix. Additionally, ClinicalTrials.gov was searched without any restrictions.

**Searching other resources.** We did not find other resources for our meta-analysis

## Data collection and analysis

**Selection of studies.** All 1117 records were exported to Refworks, duplicates were removed based on an exact match in authors, title and publication year using the duplicate removal option in Refworks. The title and abstract of the remaining 1065 records were screened by two investigators independently (LN and MA). All differences were reviewed by a third investigator (NN). 45 full articles were assessed by two investigators (LN and MA) to search for the included articles. The inclusion criteria were Randomized controlled trials, duration of 12 weeks or more, any DPPIV inhibitor on top of insulin, any diabetes duration or age. 7 were included in the study: Garg et al., 2013 [13], Georges et al., 2015 [14], Griffin et al., 2014 [15], Hari Kumar et al., 2013 [16], Wang et al., 2018 [17], Yang et al., 2021 [10], and Zhao et al., 2014 [6]. 38 articles were excluded for several reasons like duration of the study less than 3 months, population studied was not type 1 DM, the article was not a trial, duplicates, intervention not DPP4 inhibitors, comparators not on top of insulin, unfinished trials and text not in English (cf. S2 Appendix). A Prisma flow diagram summarizes the selection process. (cf. Fig 1)

The Overview of Synthesis and Included Studies table is summarized in Table 1

## Data extraction and management

Two investigators (LN) and (TN) extracted the data from the seven articles independently. All outcome related to HbA1c, glucose, C-peptide, insulin dosage, BMI, weight and adverse events were extracted. The data were entered into RevMan version 8.10.0 for analysis.

**Risk of bias assessment in included studies.** Two independent investigators (LN and TN) used the "ROB2 assessment form" for assessing the risk of bias in every study. The disagreements were resolved by a third investigator (NN). This form examines the randomization procedure, deviations from planned interventions, incomplete outcome data, outcome measurement, and choice of reported result for each outcome measured.

## Measures of treatment effect

We assessed the mean difference for continuous outcomes and the Risk Ratio for dichotomous variables.

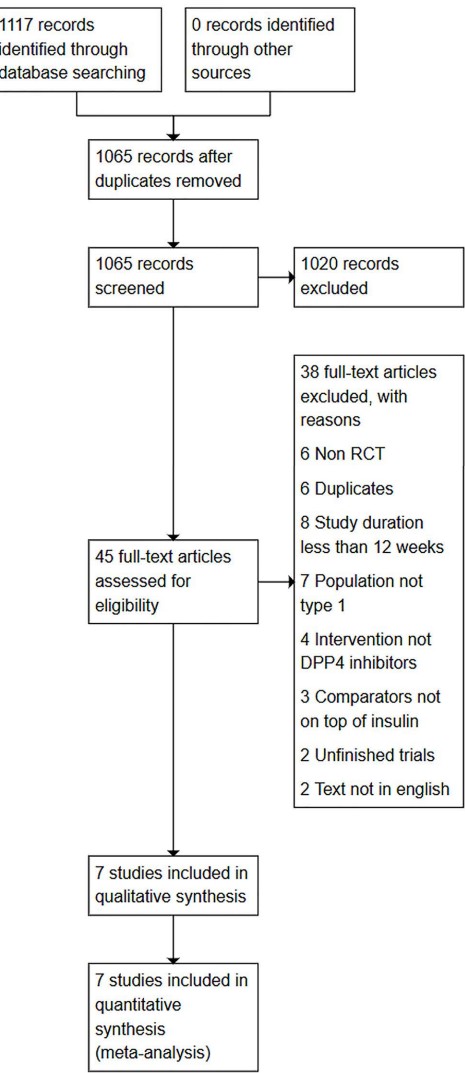

**Fig 1. Flow diagram.**

## Unit of analysis issues

HbA1c, blood glucose, and C-peptide levels were originally reported in various units across the included studies. Specifically, HbA1c was reported as a percentage (%), blood glucose in mg/dL or mmol/L, and C-peptide in ng/mL, nmol/L, or pmol/L. For consistency, HbA1c and blood glucose values were converted using an online tool available at https://www.HbA1cnet.com/HbA1c-calculator/. C-peptide levels reported in nmol/L were manually converted to pmol/L by multiplying by 1,000. Additionally, C-peptide values expressed in ng/mL or pg/mL were converted to pmol/L using an online conversion tool from UNITSLAB.COM designed for conventional to SI unit conversions.

## Dealing with missing data

Data that were represented only in graphs were extracted manually by two investigators (LN and TN), and the mean of the two values was used in the meta-analysis.

**Table 1. Overview of Synthesis and Included Studies table.**

| Study | Garg et al., 2013 | Georges et al., 2015 | Griffin et al., 2014 | Hari Kumar et al., 2013 | Yang et al., 2021 | Wang et al., 2018 | Zhao et al., 2014 |
|---|---|---|---|---|---|---|---|
| Reference | doi: 10.4158/ EP12100.OR | doi: 10.1111/ dme.13046 | doi: 10.1016/ S2213-8587(14)70115-9 | doi: 10.1016/j.dia-bres.2013.01.020 | doi: 10.1210/cli-nem/ dgab026 | doi: 10.1111/ jdi.12873 | doi: 10.1210/ jc.2013-3633 |
| Design | Multi-centered RCT | Single-centered mixed RCT | Multi-centered RCT | Single-centered RCT | Single-centered RCT | Single-centered RCT | Single-centered RCT |
| Location | USA (3 centers) | UK | USA (3 centers) | India | China | China | China |
| Experimental drug/day | Sitagliptin 100 mg + insulin | Saxagliptin 5 mg + insulin | Sitagliptin 100 mg + lansopra-zole 60 mg + insulin if adult or Sitagliptin 50 mg + lansoprazole 30 mg + insulin if <18yo | Sitagliptin 100 mg + insulin | Sitagliptin 100 mg + insulin | Sitagliptin 100 mg + insulin | Sitagliptin 100 mg + insulin |
| Comparator | Placebo + insu-lin | Pla-cebo + insulin | Placebo + insulin | Insulin | Insulin | Insulin | Insulin |
| Time points of mea-surement (months) | 4 | 3 | 3, 6, 9, 12 | 12 | 6, 12, 18, 24 | 6, 12 | 3, 6, 9, 12 |
| Experimental sample size | 63 | 14 | 46 | 6 | 22 | 20 | 15 |
| Control sam-ple size | 62 | 14 | 22 | 6 | 25 | 20 | 15 |
| Population | Type 1 diabetes | C-peptide negative Type 1 diabetes | Type 1 diabetes | Type 1 diabetes | LADA | LADA | LADA |
| Age (years) | 38 ± 14 | 42.9 ± 3.3 | 16.2 ± 5.8 | 27.2 ± 4.2 | 48.2 ± 11.8 | 49.9 ± 11.8 | 47.5 ± 3.3 |
| Diabetes duration | 21 ± 11 years | 20.5 ± 2.7 years | 103.4 ± 51.7 days | 33.4 ± 9.4 days | 2.1 ± 1.6 years | ≤ 3 years | 1.4 ± 0.2 |
| Main Outcome | HbA1c Insulin dosage Weight, BMI C-peptide, GLP-1, GIP glucagon CGM data | Hypoglycemia HbA1c insulin dosage adverse events | C-peptide glucose AUC insulin dosage HbA1c | insulin dosage C-peptide HbA1c BMI | C-peptide HbA1c BMI | insulin dosage C-peptide HbA1c glucose BMI | C-peptide HbA1c glucose |

Data is represented as mean ± Standard deviation; RCT: randomized controlled trials; USA: United States of America; UK: United Kingdom; LADA: Latent autoimmune diabetes in adults.

## Synthesis methods

The information extracted from the articles covered details such as the primary author's name, publication year, study locations and countries, trial phases, study duration, number of participants in the intervention group and the control group, participant characteristics (including age, weight, and body mass index), dosages, means and standard deviations (SDs) for both groups.

For the statistical analysis, a random-effects model was employed to aggregate the results of the included studies, considering potential heterogeneity among them. Effect sizes for continuous outcomes were represented as mean differences (MD) with 95% confidence intervals (CI) calculated by Wald-type method. The $I^2$ statistic was used to evaluate heterogeneity, with values over 50% indicating substantial heterogeneity. Sensitivity analyses were carried out to determine the

influence of individual studies on the overall results and to evaluate the robustness of the findings. All statistical analyses were performed using RevMan version 8.10.0 (www.revman.cochrane.org), unless stated otherwise.

### Investigation of heterogeneity and subgroup analysis

Study arms were categorized as follows: saxagliptin versus placebo, sitagliptin alone versus placebo, and sitagliptin combined with lansoprazole versus placebo. For analysis purposes, two groupings were created: the "sitagliptin group," which included sitagliptin with or without lansoprazole, and the broader "DPP-4 inhibitors group," which included saxagliptin as well as sitagliptin (with or without lansoprazole). Subgroup analyses were conducted based on the duration of diabetes (<1 year, 1–3 years, and >3 years), considering that the therapeutic mechanism of DPP-4 inhibitors—enhancing pancreatic β-cell function via the GLP-1 pathway [18]—may be less effective in patients with long-standing disease. Additional subgroup analyses were performed according to treatment duration (3 months, 3–6 months, and >6 months) to determine whether prolonged therapy yields improved outcomes. Finally, subgroup analysis based on BMI (<25 kg/m² versus ≥25 kg/m²) was conducted to evaluate whether overweight or obesity influences the efficacy of DPP-4 inhibitors. In the treatment duration subgroup analysis, Griffin et al (1) and Zhao et al (1) represents data taken at 3 months of treatment, Griffin et al (2), Wang et al (1), Yang et al (1) and Zhao et al (2) at 6 months of treatment.

### Assessment of reporting bias

To explore the potential influence of missing results on our findings, we qualitatively evaluated reporting bias across the included studies. While no formal funnel plot was feasible due to the use of Revman software, we examined outcome reporting consistency against trial registrations and protocols when available. Most studies included clearly predefined outcomes and comprehensive result reporting; however, a few exhibited incomplete outcome disclosure, raising a modest concern for selective reporting bias. Although no significant asymmetry was observed in the limited meta-analytic comparisons, we acknowledge that publication and reporting bias cannot be fully excluded.

### Assessment of certainty in the body of evidence

While we did not apply a formal GRADE framework, we informally assessed the overall certainty of the evidence by considering study design, consistency, precision, and directness across outcomes. The evidence supporting the reduction in insulin requirements and postprandial glucose was derived from randomized trials with mostly low to moderate risk of bias and showed consistent directionality across subgroups. However, due to the small sample size, occasional imprecision, and some concerns over risk of bias in a subset of studies, we consider the overall confidence in the effect estimates to range from moderate to low depending on the specific outcome.

## Results

### Risk of bias of included studies

The methodological quality of the included randomized controlled trials was assessed using the Cochrane Risk of Bias 2 (RoB 2) tool, which evaluates five domains: the randomization process, deviations from the intended interventions, missing outcome data, measurement of the outcome, and selection of the reported result. Among the seven studies assessed, three (Yang et al., 2021; Garg et al., 2013; and Griffin et al., 2014) were judged to have an overall low risk of bias across all domains. Four studies (Geroge et al., 2015; Zhao et al., 2014; Kumar et al., 2013; and Wang et al., 2018) were rated as having some concerns overall. Notably, Kumar et al., 2013 exhibited a high risk of bias in two key domains: the randomization process and deviations from intended interventions. The most commonly encountered concerns across studies were related to the randomization process and deviations from the intended interventions. These findings suggest

that, while the majority of included studies were of acceptable methodological quality, some limitations in study design or reporting may have introduced potential bias. (Fig 2)

## HbA1c levels

There was no significant difference in HbA1c levels between the DPP-4 inhibitor and placebo groups (mean difference [MD] = 0.03; 95% confidence interval [CI]: −0.05 to 0.11; p = 0.48) (Fig 3). However, in subgroup analyses based on treatment duration, HbA1c was significantly higher with DPP-4 inhibitors at 3 months (MD = 0.23; 95% CI: 0.05 to 0.41; p = 0.01). Conversely, between 3–6 months of treatment, HbA1c was significantly lower with DPP-4 inhibitors compared to placebo (MD = −0.10; 95% CI: −0.16 to −0.05; p = 0.0003). No significant difference was observed with treatment durations exceeding 6 months (MD = 0.03; 95% CI: −0.06 to 0.12; p = 0.57) (S2 Appendix). There was no significant difference in HbA1c changes between both groups in other subgroup analysis: for BMI subgroup analysis: MD = 0.03 [−0.06, 0.12], p = 0.57 for BMI < 25 kg/m2 and MD = 0.05 [−0.20, 0.30], p = 0.7 for BMI > 25 kg/m2 (S2 Appendix); for diabetes onset

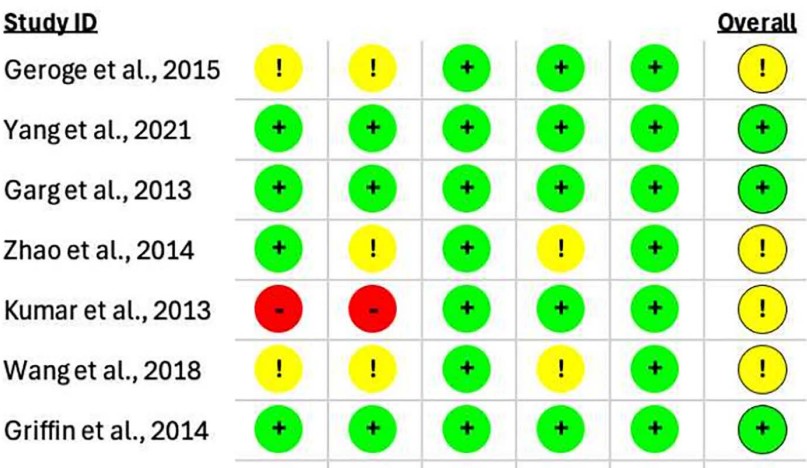

**Fig 2. Risk of bias.**

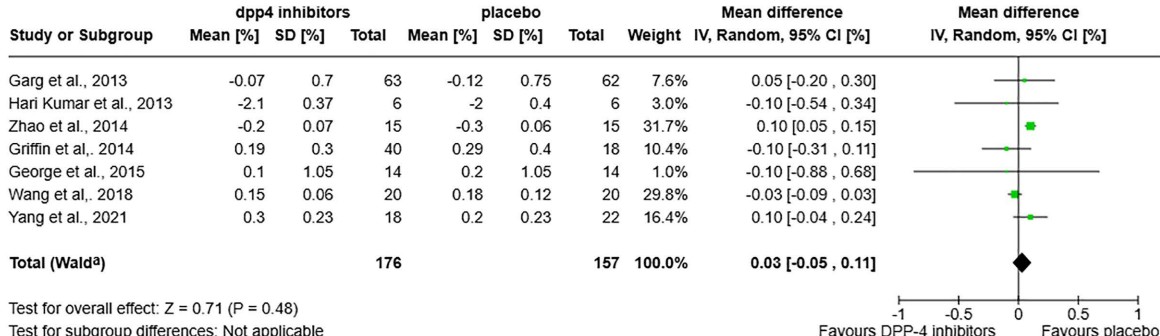

| Study or Subgroup | dpp4 inhibitors | | | placebo | | | Weight | Mean difference IV, Random, 95% CI [%] | Mean difference IV, Random, 95% CI [%] |
|---|---|---|---|---|---|---|---|---|---|
| | Mean [%] | SD [%] | Total | Mean [%] | SD [%] | Total | | | |
| Garg et al., 2013 | -0.07 | 0.7 | 63 | -0.12 | 0.75 | 62 | 7.6% | 0.05 [-0.20 , 0.30] | |
| Hari Kumar et al., 2013 | -2.1 | 0.37 | 6 | -2 | 0.4 | 6 | 3.0% | -0.10 [-0.54 , 0.34] | |
| Zhao et al., 2014 | -0.2 | 0.07 | 15 | -0.3 | 0.06 | 15 | 31.7% | 0.10 [0.05 , 0.15] | |
| Griffin et al,. 2014 | 0.19 | 0.3 | 40 | 0.29 | 0.4 | 18 | 10.4% | -0.10 [-0.31 , 0.11] | |
| George et al., 2015 | 0.1 | 1.05 | 14 | 0.2 | 1.05 | 14 | 1.0% | -0.10 [-0.88 , 0.68] | |
| Wang et al,. 2018 | 0.15 | 0.06 | 20 | 0.18 | 0.12 | 20 | 29.8% | -0.03 [-0.09 , 0.03] | |
| Yang et al., 2021 | 0.3 | 0.23 | 18 | 0.2 | 0.23 | 22 | 16.4% | 0.10 [-0.04 , 0.24] | |
| **Total (Wald[a])** | | | 176 | | | 157 | 100.0% | 0.03 [-0.05 , 0.11] | |

Test for overall effect: Z = 0.71 (P = 0.48)
Test for subgroup differences: Not applicable
Heterogeneity: Tau² (DL[b]) = 0.00; Chi² = 14.58, df = 6 (P = 0.02); I² = 59%

**Footnotes**
[a]CI calculated by Wald-type method.
[b]Tau² calculated by DerSimonian and Laird method.

**Fig 3. Forest plot for HbA1c in the DPP-4 inhibitors vs. placebo group.**

subgroups: MD = −0.10 [−0.29, 0.09] p = 0.29 for diabetics <1 year, MD = 0.05 [−0.05, 0.15] p = 0.32 for diabetics for 1 to 3 years and 0.04 [−0.21, 0.28], p = 0.77 for diabetics >3 years (S2 Appendix).

In the comparison between the sitagliptin (with or without lansoprazole) arm and the placebo arm, no significant difference in HbA1c changes was observed between the two groups (MD = 0.03; 95% CI: −0.05 to 0.11; p = 0.48) (S2 Appendix). Subgroup analyses yielded similar results to those observed in the overall DPP-4 inhibitors arm. For treatment duration, HbA1c was significantly higher with sitagliptin at 3 months (MD = 0.25; 95% CI: 0.06 to 0.44; p = 0.009), significantly lower between 3 to 6 months (MD = −0.10; 95% CI: −0.16 to −0.05; p = 0.0003), and showed no significant difference beyond 6 months (MD = 0.03; 95% CI: −0.06 to 0.12; p = 0.57) (S2 Appendix). In the BMI subgroup analysis, no significant differences were observed for patients with BMI < 25 kg/m² (MD = 0.03; 95% CI: −0.06 to 0.12; p = 0.57) or BMI ≥ 25 kg/m² (MD = 0.05; 95% CI: −0.20 to 0.30; p = 0.70) (S2 Appendix). Regarding diabetes onset, there were no significant differences in patients with diabetes duration <1 year (MD = −0.10; 95% CI: −0.29 to 0.09; p = 0.29), 1–3 years (MD = 0.05; 95% CI: −0.05 to 0.15; p = 0.32), or >3 years (MD = 0.05; 95% CI: −0.20 to 0.30; p = 0.70) (S2 Appendix).

In the comparison between the sitagliptin group and the placebo group, no significant difference in HbA1c changes was observed (MD = 0.05; 95% CI: −0.04 to 0.13; p = 0.30) (S2 Appendix). In subgroup analyses by treatment duration, HbA1c was significantly lower with sitagliptin compared to placebo between 3 to 6 months of treatment (MD = −0.11; 95% CI: −0.17 to −0.06; p < 0.0001), whereas no significant difference was observed beyond 6 months (MD = 0.04; 95% CI: −0.05 to 0.14; p = 0.36) (S2 Appendix:). No significant differences were observed in other subgroup analyses. In the BMI subgroup analysis, the mean difference was 0.04 (95% CI: −0.05 to 0.14; p = 0.36) for patients with BMI < 25 kg/m², and 0.05 (95% CI: −0.20 to 0.30; p = 0.70) for those with BMI ≥ 25 kg/m² (S2 Appendix). In the diabetes onset subgroups, there was no significant difference for patients with diabetes duration <1 year (MD = −0.10; 95% CI: −0.54 to 0.34; p = 0.65), 1–3 years (MD = 0.05; 95% CI: −0.05 to 0.15; p = 0.32), or >3 years (MD = 0.05; 95% CI: −0.20 to 0.30; p = 0.70) (S2 Appendix).

### C-peptide

There was no significant difference between DPP-4 inhibitors and placebo concerning fasting C-peptide: MD = 35.05 [−59.79, 129.89], p = 0.47 (S2 Appendix), subgroup analysis regarding study duration was also performed and showed no significant difference except in the 3 months treatment, where the fasting C peptide was significantly higher with DPP-4 inhibitors: MD = 107.20 [92.42, 121.98], p < 0.00001 for 3 months of treatment, MD = −16.47 [−138.28, 105.34], p = 0.79 between 3 to 6 months of treatment, MD = 35.05 [−59.79, 129.89] p = 0.47 for >6 months of treatment (S2 Appendix).

Post prandial C-peptide was higher with DPP-4 inhibitors vs. placebo but the difference was non-significant: MD = 213.74 [−39.92, 467.40], p = 0.1 (S2 Appendix). In a subgroup analysis according to treatment duration, post prandial C-peptide was significantly higher with DPP-4 inhibitors vs. placebo at 3 months of treatment: MD = 342.90 [214.51, 471.29], p < 0.00001, however, there were no difference in the other subgroups: MD = 111.90 [−159.34, 383.14], p = 0.42 in 3 to 6 months, MD = 213.74 [−39.92, 467.40], p = 0.1 in >6 months (S2 Appendix). Post prandial C-peptide was non-significantly higher with DPP-4 inhibitors vs. placebo in the subgroup analysis according to diabetes duration: MD = 19.91 [−11.97, 51.79], p = 0.22 in diabetics < 1 year vs. MD = 279.89 [−7.78, 567.57], p = 0.06 for diabetics since 1 to 3 years ago (S2 Appendix).

Concerning ΔC-peptide, it was non-significantly higher with DPP-4 inhibitors vs. placebo: MD = 266.61 [−50.67, 583.88], p = 0.1 (S2 Appendix). The analysis conducted according to treatment duration showed a significant increase in ΔC-peptide in DPP-4 arm vs. placebo at 3 months and between 3 to 6 months of treatment: MD = 340.50 [293.07, 387.93], p < 0.00001 and MD = 141.85 [5.02, 278.67], p = 0.04, respectively. However, no difference was found in the other subgroup: MD = 266.61 [−50.67, 583.88], p = 0.1 for > 6 months of treatment (S2 Appendix).

There was no difference in the mean difference between DPP-4 inhibitors and placebo concerning C-peptide AUC of 2 and 4 hours post prandial: MD = 30.50 [−34.35, 95.35], p = 0.36 (S2 Appendix:) and MD = 0.02 [−1.11, 1.15], p = 0.97 respectively (S2 Appendix).

## Insulin dosage

Concerning the total daily insulin dosage, there was a significant reduction by −3.56 [−5.40, −1.73] units per day with DPP-4 inhibitors vs. placebo (P = 0.0001) (Fig 4). This reduction was also seen in the subgroup analysis of treatment time periods. (S2 Appendix). However, this reduction was only seen in patients with a normal BMI (<25 Kg/m2): MD = −3.80 [−5.98, −1.61], p = 0.0007 but no significant effect was seen in BMI > 25 Kg/m$^2$ MD = −2.26 [−5.89, 1.37], p = 0.22 (S2 Appendix). As for diabetes duration, a significant reduction in insulin units was seen only in patients with <3 years of diabetes: MD = −8.30 [−15.85, −0.75] p = 0.03 (for <1 year of diabetes, MD = −3.45 [−5.70, −1.20], p = 0.003; for 1–3 years post diagnosis vs. MD = −2.77 [−5.83, 0.29], p = 0.08; for > 3 years post diagnosis) (S2 Appendix). The reduction in insulin dosages was also seen in the Sitagliptin analysis and paralleled the above results (S2 Appendix)

For basal insulin units, there was no significant difference between the DPP-4 inhibitors and the placebo: MD = −0.39 [−1.85, 1.06], p = 0.6 (S2 Appendix). As for bolus insulin units there is a significant reduction of 2.61 units per day [−4.89, −0.32], p = 0.03 with DPP-4 inhibitors vs. placebo (S2 Appendix).

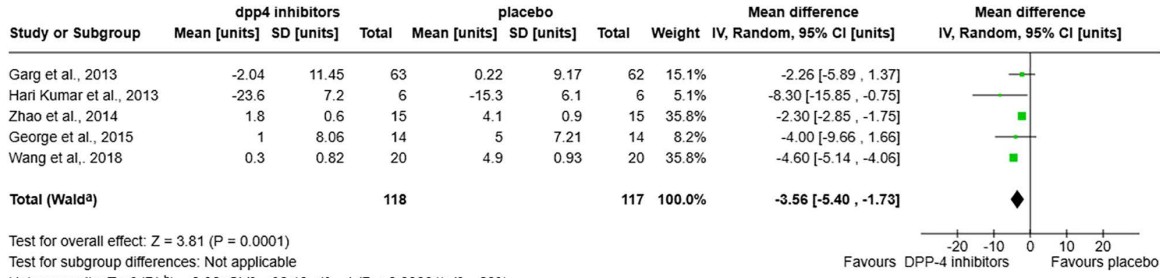

**Fig 4. Forest plot for insulin units/day in the DPP-4 inhibitors vs. placebo group.**

Concerning insulin dosage adjusted for weight, there is no difference between the DPP-4 and placebo group MD = 0.00 [−0.07, 0.07], p = 0.92 (S2 Appendix).

## Glucose

Zhao et al., 2014 and Wang et al., 2018 studied fasting and post prandial blood glucose and there were no significant difference in the DPP-4 arm vs. placebo in our meta-analysis for fasting blood glucose, however, post prandial blood glucose was significantly lower with DPP-4 vs. placebo (P < 0.00001). Garg et al, 2013 assessed Δ blood glucose, CGM data. Δ blood glucose was significantly lower with DPP-4 inhibitors vs. placebo (p < 0.00001) but there were no significant difference between the 2 arms concerning CGM data: time in range, time in hyperglycemia, time in hypoglycemia and mean glucose: p = 0.67, p = 0.28, p = 0.08 and p = 0.61 respectively. Griffin et al., 2014 studied glucose area under the curve (AUC) but there was no difference between the 2 groups (p = 0.28).

## BMI and weight

Garg et al., Hari Kumar et al., Zhao et al., Wang et al., and Yang et al. evaluated the impact of DPP-4 inhibitors compared to placebo on BMI in their respective studies. In our meta-analysis, no significant difference was observed (p = 0.18), and subgroup analyses similarly revealed no differences.

Garg et al., Hari Kumar et al., and Zhao et al. evaluated the effect of DPP-4 inhibitors versus placebo on weight. No significant difference in weight (kg) was observed between the DPP-4 inhibitor group and the placebo group (p = 0.4), and subgroup analyses similarly showed no significant differences.

## Other outcomes

Yang et al 2021 studied HOMA2-b score and the difference was higher in the DPP-4 group vs. placebo: (p = 0.02).

## Adverse events

DPP-4 inhibitors appeared safe in patients with type 1 diabetes compared to placebo (risk ratio [RR] = 1.00; 95% CI: 0.77–1.31; p = 0.98). No significant differences in adverse events were observed in subgroup analyses: RR = 0.49 (95% CI: 0.09–2.59; p = 0.40) for treatment durations of 3–6 months, and RR = 1.19 (95% CI: 0.50–2.84; p = 0.69) for treatment durations longer than 6 months. Similarly, no significant differences were found when stratified by BMI: RR = 1.19 (95% CI: 0.50–2.84; p = 0.69) for BMI < 25 kg/m², and RR = 0.49 (95% CI: 0.09–2.59; p = 0.40) for BMI ≥ 25 kg/m². Regarding diabetes duration, no significant differences in adverse events were detected: RR = 1.00 (95% CI: 0.90–1.12; p = 0.97) for patients with diabetes duration <1 year, RR = 3.67 (95% CI: 0.42–32.30; p = 0.24) for 1–3 years, and RR = 0.49 (95% CI: 0.09–2.59; p = 0.40) for more than 3 years. In the sitagliptin-only subgroup, no significant difference in adverse events was observed compared to placebo (RR = 1.18; 95% CI: 0.17–8.32; p = 0.87). Subgroup analysis based on study duration similarly revealed no significant differences: RR = 0.49 (95% CI: 0.09–2.59; p = 0.40) for 3–6 months of treatment and RR = 3.67 (95% CI: 0.42–32.30; p = 0.24) for treatment durations greater than 6 months.

## Discussion

This meta-analysis evaluated the impact of adding DPP-4 inhibitors to insulin therapy, focusing on glycemic control as assessed by HbA1c, insulin requirements, and the implications for patients with type 1 diabetes. The pooled data demonstrated that the addition of DPP-4 inhibitors did not result in a significant or sustained reduction in HbA1c levels, except within a narrow treatment window of 3 to 6 months. This transient effect suggests that while DPP-4 inhibitors may offer short-term glycemic benefits, these are not maintained with longer use in patients already on insulin therapy.

A consistent finding across the included studies was a modest yet statistically significant reduction in daily insulin requirements. This insulin-sparing effect may be attributed to improved postprandial glucose control or enhanced endogenous incretin activity, even in patients already receiving insulin therapy. Glucagon-like peptide-1 (GLP-1) is an incretin hormone that promotes insulin secretion in response to elevated blood glucose levels, suppresses glucagon secretion, delays gastric emptying, and decreases caloric intake, likely through central nervous system signaling [19]. GLP-1 is produced by intestinal L cells through cleavage of pro-glucagon and exists in two active forms: GLP-1(7−36) and GLP-1(7−37) [20]. Dipeptidyl peptidase-4 (DPP-4), secreted by endothelial cells, rapidly inactivates GLP-1 by cleaving its two N-terminal amino acids, thereby reducing its biological activity [21,22]. Inhibiting DPP-4 leads to increased circulating GLP-1 levels, resulting in greater endogenous insulin secretion and lower postprandial glucose concentrations [23]. This mechanism likely explains the observed reduction in daily insulin requirements with DPP-4 inhibitor therapy. Additionally, it raises the possibility that DPP-4 inhibitors may enhance insulin sensitivity or reduce glucotoxicity in specific subgroups possibly the overweight group, potentially helping to mitigate complications associated with high-dose insulin use.

While traditionally considered a disease of absolute insulin deficiency, emerging evidence supports the presence of insulin resistance in a subset of type 1 patients—particularly those with longer disease duration, obesity, or metabolic syndrome features. In these individuals, DPP-4 inhibitors may offer additional metabolic benefits beyond glycemic control, including potential effects in β-cell response and insulin sensitivity, although these remain to be fully elucidated.

DPP-4 inhibitors did not improve weight, BMI, fasting blood glucose, glucose AUC, time in range, time in hyperglycemia, time in hypoglycemia or mean glucose. DPP-4 inhibitors are known to be weight neutral, thus, a non-effect versus placebo was expected on weight and BMI [24]. DPP-4 inhibitors also did not improve fasting C-peptide and post prandial C-peptide except for 3 months of treatment for fasting C- peptide and or post prandial C peptide suggesting a better beta cell function on the short term.

Although there has been no difference with DPP-4 inhibitors on fasting blood glucose, we have a lower post prandial blood glucose by 34.40 mg/dL in patients treated with DPP-4 inhibitors vs. placebo especially in patients with a BMI less than 25 kg/m$^2$ and in new onset diabetes <3 years. In regards of the daily insulin dosage adjusted for weight there was a non-significant difference between the two arms. The bolus insulin units were reduced with DPP-4 compared to placebo and the basal insulin doses were the same between the two groups suggesting possible improvement in insulin resistance. Regarding β-cell function, the HOMA2-b was higher in the DPP-4 group by 31.60% suggesting an increased β-cell function. Concerning adverse events, DPP-4 inhibitors seem safe compared with placebo.

## Limitations

Georges et al. used a mixed RCT trial model that can cause some bias in the results, the total number of studies and participants is small hindering an accurate subgroup analysis; thus, further studies are needed to generate more robust data.

Two articles were not included because they are not in English (Fengyi et al., 2015 [25] and Luo et al., 2015 [26]) although they might have contained important data for our analysis. Data extracted manually from the graphs might differ from the original data from the article. The analysis of adverse events included all reported events, without distinction based on their rarity or severity. In addition the included trials are small RCTs and this may limit the generalizability of our results to the general population.

## Agreements and disagreements with other studies or reviews

Previous meta-analyses and individual studies have shown no significant effect of DPP-4 inhibitors compared to placebo on HbA1c levels. However, some studies have suggested potential benefits on postprandial C-peptide [6,10] and postprandial glucose levels [17].

## Conclusion

In conclusion, DPP-4 inhibitors appear to be safe as adjunctive therapy to insulin in patients with type 1 diabetes. Although no significant long-term benefit on HbA1c reduction was observed, these agents may help reduce daily insulin requirements, particularly bolus doses, lower postprandial glucose levels, and transiently increase postprandial C-peptide concentrations.

Overall, while DPP-4 inhibitors do not seem to significantly lower HbA1c in insulin-treated patients over the long term, their potential role in improving insulin dynamics and reducing insulin needs, particularly among insulin-resistant phenotypes, warrants further investigation. Future studies should focus on identifying subgroups most likely to benefit and exploring whether earlier or intermittent use of DPP-4 inhibitors could sustain glycemic improvements.

Additional large-scale, long-term studies are needed to better define the impact of DPP-4 inhibitors on HbA1c and β-cell function in patients with type 1 diabetes.

## Supporting information

**S1. Prisma 2020 Checklist.**
(DOCX)

**S2 Appendix. Appendix.**
(DOCX)

## Acknowledgments

The authors would like to thank the University of Balamand and its faculty of Medicine.

## Author contributions

**Conceptualization:** Nancy Fawzi Nakhoul.

**Data curation:** Laurette Nakhoul, Tracy Nakhoul.

**Formal analysis:** Laurette Nakhoul, Tracy Nakhoul, Frederic Harb.

**Investigation:** Laurette Nakhoul, Maria Abi Azar, Nancy Fawzi Nakhoul.

**Methodology:** Frederic Harb, Nancy Fawzi Nakhoul.

**Project administration:** Nancy Fawzi Nakhoul.

**Resources:** Laurette Nakhoul, Maria Abi Azar.

**Software:** Laurette Nakhoul, Frederic Harb.

**Supervision:** Frederic Harb, Nancy Fawzi Nakhoul.

**Validation:** Laurette Nakhoul, Tracy Nakhoul, Frederic Harb, Nancy Fawzi Nakhoul.

**Visualization:** Laurette Nakhoul, Frederic Harb.

**Writing – original draft:** Laurette Nakhoul, Tracy Nakhoul.

**Writing – review & editing:** Frederic Harb, Nancy Fawzi Nakhoul.

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
