## [Decision Letter · Decision Letter 0]

12 Jun 2025

PONE-D-25-24150The efficacy of DPP IV inhibitors as adjunct therapy for patients with auto-immune Diabetes: A Systematic review and Meta-analysisPLOS ONE

Dear Dr. Nakhoul,

Thank you for submitting your manuscript to PLOS ONE. After careful consideration, we feel that it has merit but does not fully meet PLOS ONE’s publication criteria as it currently stands. Therefore, we invite you to submit a revised version of the manuscript that addresses the points raised during the review process.

We look forward to receiving your revised manuscript.

Kind regards,

Toshiki Maeda

Academic Editor

PLOS ONE

Journal Requirements:

4. Please remove all personal information, ensure that the data shared are in accordance with participant consent, and re-upload a fully anonymized data set.

Reviewers' comments:

Reviewer's Responses to Questions

**Comments to the Author**

1. Is the manuscript technically sound, and do the data support the conclusions?

Reviewer #1: Yes

Reviewer #2: Partly

2. Has the statistical analysis been performed appropriately and rigorously? 

Reviewer #1: Yes

Reviewer #2: Yes

3. Have the authors made all data underlying the findings in their manuscript fully available?

Reviewer #1: Yes

Reviewer #2: Yes

4. Is the manuscript presented in an intelligible fashion and written in standard English?

Reviewer #1: Yes

Reviewer #2: Yes

5. Review Comments to the Author

Reviewer #1: Dear Authors,

I am pleased to have the opportunity to review your manuscript. Your systematic review and meta-analysis address an interesting and highly debated clinical question regarding the efficacy and safety of DPP-4 inhibitors as an adjunct therapy for patients with type 1 diabetes mellitus (T1DM). Your methodology largely follows standard practices, and your comprehensive search strategy and quality assessment using the Cochrane Risk of Bias Tool 2.0 are commendable. However, there seem to be several significant problems with your study.

1. In many countries, DPP-4 inhibitors are not approved for use in patients with T1DM. In this context, studies involving the administration of these drugs to patients with T1DM may represent a specialized target population, and their validity should be substantiated. It is also important to acknowledge that the use of DPP-4 inhibitors in T1DM patients is often off-label and not covered by current guidelines or reimbursement policies in many countries. The included RCTs may have been conducted in specific research settings or may have involved a limited patient population (e.g., patients with early T1DM and preserved beta-cell function or purely exploratory studies conducted before regulatory approvals). More detailed information and discussion are needed regarding the background of these RCTs. This would clarify the extent to which your findings apply to routine clinical practice and highlight the limitations of generalizability.

2. Although the limitations section mentions the study duration, it is unclear how long-term administration would affect pancreatic beta-cell survival. Therefore, stating that this therapy is "to preserve residual beta-cell function" may not be entirely accurate. Either provide justification to explain this clearly to the reader, or consider modifying the wording in your response. Several times in your discussion and conclusion, you state that an increase in C-peptide levels suggests "the possibility of preserving or improving residual β-cell function." Ultimately, you conclude that DPP-4 inhibitors are "a safe and effective adjunct therapy to preserve residual β-cell function." While an increase in C-peptide may indicate an improvement in beta-cell insulin secretory response, it cannot be confirmed whether this directly translates to "survival" or "preservation" without much longer-term follow-up studies. This wording might mislead readers. I recommend using more cautious language. Alternatively, if you believe "preservation" is justified, please provide a more detailed discussion of the underlying physiological mechanisms or existing evidence from longer-term studies that support this claim.³

3. Although a significant reduction in the total daily insulin dose (MD = -2.5 units/day) is reported, the clinical significance of this reduction in the daily lives of patients with type 1 diabetes mellitus (T1DM) should be discussed more specifically. For example, does this translate to fewer injections, reduced risk of weight gain, or reduced risk of hypoglycemia? Discussing the implications for patients' quality of life or management burden would enhance the clinical relevance of this finding.

Reviewer #2: Definition and Justification of Risk and Treatment Burden

Please provide a clearer explanation of how the “10-year cardiovascular risk” is calculated, including whether any model calibration was performed for individual countries.

Define how "treatment burden" is operationalized—does it include polypharmacy, cost, frequency of monitoring, etc.?

Statistical Transparency

Clarify if complex survey designs (e.g., stratification, clustering) were adjusted for in the confidence intervals.

Provide more details on how survey weights were applied.

If multivariable modeling was considered and ruled out, please explain why.

Overgeneralization of Conclusions

The current wording implies causal conclusions in some sections. Please revise to maintain descriptive tone.

Statements such as “policy should prioritize targeting high-risk individuals” are reasonable but should be supported with more robust discussion of feasibility, health system capacity, and heterogeneity between LMICs.

Country Comparisons

Consider stratifying countries by World Bank income classification or region for comparative clarity.

Adding a figure to visualize between-country differences in risk or burden would improve accessibility.

6. PLOS authors have the option to publish the peer review history of their article (what does this mean? ). If published, this will include your full peer review and any attached files.

**Do you want your identity to be public for this peer review?** For information about this choice, including consent withdrawal, please see our Privacy Policy .

Reviewer #1: No

Reviewer #2: No

---

## [Author Response · Author response to Decision Letter 1]

27 Jul 2025

Reviewer #1: Dear Authors,

I am pleased to have the opportunity to review your manuscript. Your systematic review and meta-analysis address an interesting and highly debated clinical question regarding the efficacy and safety of DPP-4 inhibitors as an adjunct therapy for patients with type 1 diabetes mellitus (T1DM). Your methodology largely follows standard practices, and your comprehensive search strategy and quality assessment using the Cochrane Risk of Bias Tool 2.0 are commendable. However, there seem to be several significant problems with your study.

1. In many countries, DPP-4 inhibitors are not approved for use in patients with T1DM. In this context, studies involving the administration of these drugs to patients with T1DM may represent a specialized target population, and their validity should be substantiated. It is also important to acknowledge that the use of DPP-4 inhibitors in T1DM patients is often off-label and not covered by current guidelines or reimbursement policies in many countries. The included RCTs may have been conducted in specific research settings or may have involved a limited patient population (e.g., patients with early T1DM and preserved beta-cell function or purely exploratory studies conducted before regulatory approvals). More detailed information and discussion are needed regarding the background of these RCTs. This would clarify the extent to which your findings apply to routine clinical practice and highlight the limitations of generalizability.

Thank you for your comments, we have updated the manuscript to clarify the off label use of DPP-4 in T!DM. we have updated the discussion to include the background of these studies when mentioned, and added the limitation o generalizability in the discussion .

2. Although the limitations section mentions the study duration, it is unclear how long-term administration would affect pancreatic beta-cell survival. Therefore, stating that this therapy is "to preserve residual beta-cell function" may not be entirely accurate. Either provide justification to explain this clearly to the reader, or consider modifying the wording in your response. Several times in your discussion and conclusion, you state that an increase in C-peptide levels suggests "the possibility of preserving or improving residual β-cell function." Ultimately, you conclude that DPP-4 inhibitors are "a safe and effective adjunct therapy to preserve residual β-cell function." While an increase in C-peptide may indicate an improvement in beta-cell insulin secretory response, it cannot be confirmed whether this directly translates to "survival" or "preservation" without much longer-term follow-up studies. This wording might mislead readers. I recommend using more cautious language. Alternatively, if you believe "preservation" is justified, please provide a more detailed discussion of the underlying physiological mechanisms or existing evidence from longer-term studies that support this claim.

We Thank the reviewer for this comment and we have revised the manuscript to use more cautious language.Specifically we have replaced the word preserve/preservation by to potential improvement.

3. Although a significant reduction in the total daily insulin dose (MD = -2.5 units/day) is reported, the clinical significance of this reduction in the daily lives of patients with type 1 diabetes mellitus (T1DM) should be discussed more specifically. For example, does this translate to fewer injections, reduced risk of weight gain, or reduced risk of hypoglycemia? Discussing the implications for patients' quality of life or management burden would enhance the clinical relevance of this finding.

We thank the reviewer for his valuable comment. Clinically, a reduction of approximately 3-4 insulin units per day can be meaningful for several reasons:

- Reduced injection burden: While the absolute reduction of ~3 units per day may not necessarily lead to fewer injections in patients using multiple daily injection regimens, it may lead to a lower requirement for bolus insulin. Indeed, the meta-analysis found that the most pronounced insulin-sparing effect was specifically on bolus insulin units (-2.61 units/day). This decrease could facilitate simplified carbohydrate counting and more flexible meal planning.

- Hypoglycemia risk reduction: Although the study did not directly show significant changes in hypoglycemia events, reduced insulin doses, particularly bolus insulin, theoretically lower hypoglycemia risk by minimizing insulin excess during periods of unpredictable carbohydrate intake or physical activity. This is particularly relevant for patients with variable daily routines or dietary habits.

- Weight management: Despite the lack of direct impact on weight and BMI observed in this analysis, insulin reduction, particularly bolus insulin, may indirectly prevent insulin-associated weight gain. Lower insulin doses could mitigate insulin-driven lipogenesis, contributing positively to weight management strategies in the longer term.

- Quality of life and patient satisfaction: Even modest insulin reductions can positively influence patient perceptions regarding the complexity of diabetes management. Reduced insulin dosing can ease mental burdens related to meticulous dosage calculations, thereby improving adherence and overall satisfaction with diabetes care.

Thus, although the numerical change might appear modest, the broader implications regarding patient management, potential hypoglycemia prevention, and quality of life enhancements underscore the clinical relevance of these findings.

Reviewer #2: Definition and Justification of Risk and Treatment Burden

Please provide a clearer explanation of how the “10-year cardiovascular risk” is calculated, including whether any model calibration was performed for individual countries.

Thank You for Your comment , we feel the comment does not belong to our manuscript since we did not include any the “10-year cardiovascular risk” calculation in our manuscript.

Define how "treatment burden" is operationalized—does it include polypharmacy, cost, frequency of monitoring, etc.?

Thank You for Your comment , we feel the comment does not belong to our manuscript since we did not include any the “ Treatment Biurden” calculation in our manuscript.

Statistical Transparency

Clarify if complex survey designs (e.g., stratification, clustering) were adjusted for in the confidence intervals.

We thank the reviewer for his relevant question. The meta-analysis presented was based on randomized controlled trials (RCTs), and it explicitly used a random-effects model to pool results due to expected clinical and methodological heterogeneity. This statistical approach inherently considers between-study variance. However, the meta-analysis did not explicitly indicate the adjustment for complex survey designs such as stratification or clustering because the included studies were individual RCTs with standard randomization procedures rather than observational or epidemiological surveys.

RCTs typically do not employ complex sampling designs that require adjustments (such as stratification or clustering) seen in population-based surveys. Therefore, adjustments specifically for complex survey designs were not applicable or reported in this meta-analysis.

Provide more details on how survey weights were applied.

We thank the reviewer for this constructive remark. The current meta-analysis included randomized controlled trials that inherently do not apply survey weights typically used in observational studies or population-based surveys. Survey weighting is relevant primarily when aiming to ensure the representativeness of results to broader populations or accounting for sampling methods in cross-sectional, observational studies.

Since the meta-analysis combined data from RCTs with equal weighting through a random-effects approach, the typical survey weighting methods (such as those applied in observational epidemiology) were neither applicable nor required.

If multivariable modeling was considered and ruled out, please explain why.

We thank the reviewer for his valuable comment. Multivariable modeling is generally employed in observational studies or epidemiological analyses where adjustment for potential confounding variables is crucial. In contrast, meta-analyses of randomized controlled trials inherently aim to minimize confounding through randomization. Therefore, standard meta-analytic practice focuses on direct comparisons using aggregated unadjusted data, typically without multivariable modeling.

Specifically, in this meta-analysis:

Randomization: RCT designs included in this review provided inherent control for confounding variables, limiting the necessity for multivariable adjustments.

Heterogeneity management: Instead of multivariable modeling, subgroup analyses (by treatment duration, diabetes duration, BMI) and sensitivity analyses were explicitly performed to explore heterogeneity and robustness of the results, adequately managing potential variability.

Hence, multivariable modeling was appropriately not employed given the context of the study design (meta-analysis of randomized trials) and the statistical methodology chosen.

Overgeneralization of Conclusions

The current wording implies causal conclusions in some sections. Please revise to maintain descriptive tone.

Statements such as “policy should prioritize targeting high-risk individuals” are reasonable but should be supported with more robust discussion of feasibility, health system capacity, and heterogeneity between LMICs.

Country Comparisons

Consider stratifying countries by World Bank income classification or region for comparative clarity.

We thank the reviewer for this valuable suggestion. We agree that stratifying the included studies by World Bank income classification will enhance comparative clarity. Accordingly, we will include this stratification if this comment is related to our manuscript.

Adding a figure to visualize between-country differences in risk or burden would improve accessibility.

---

## [Decision Letter · Decision Letter 1]

28 Aug 2025

The efficacy of DPP IV inhibitors as adjunct therapy for patients with auto-immune Diabetes: A Systematic review and Meta-analysis

PONE-D-25-24150R1

Dear Dr. Nakhoul,

We’re pleased to inform you that your manuscript has been judged scientifically suitable for publication and will be formally accepted for publication once it meets all outstanding technical requirements.

Kind regards,

Toshiki Maeda

Academic Editor

PLOS ONE

Additional Editor Comments (optional):

Reviewer #1:

Reviewers' comments:

Reviewer's Responses to Questions

**Comments to the Author**

1. If the authors have adequately addressed your comments raised in a previous round of review and you feel that this manuscript is now acceptable for publication, you may indicate that here to bypass the “Comments to the Author” section, enter your conflict of interest statement in the “Confidential to Editor” section, and submit your "Accept" recommendation.

Reviewer #1: All comments have been addressed

2. Is the manuscript technically sound, and do the data support the conclusions?

Reviewer #1: Yes

3. Has the statistical analysis been performed appropriately and rigorously? 

Reviewer #1: Yes

4. Have the authors made all data underlying the findings in their manuscript fully available?

Reviewer #1: Yes

5. Is the manuscript presented in an intelligible fashion and written in standard English?

Reviewer #1: Yes

6. Review Comments to the Author

Reviewer #1: Dear Authors,

Thank you for your detailed and thoughtful response. Your answer completely clarifies my questions.

I have no further comments and questions.

7. PLOS authors have the option to publish the peer review history of their article (what does this mean? ). If published, this will include your full peer review and any attached files.

**Do you want your identity to be public for this peer review?** For information about this choice, including consent withdrawal, please see our Privacy Policy .

Reviewer #1: No

---

## [Editor Report · Acceptance letter]

PONE-D-25-24150R1

PLOS ONE

Dear Dr. Nakhoul,

I'm pleased to inform you that your manuscript has been deemed suitable for publication in PLOS ONE. Congratulations! Your manuscript is now being handed over to our production team.

Kind regards,

on behalf of

Dr. Toshiki Maeda

Academic Editor

PLOS ONE